# Scalant Removal at Acidic pH for Maximum Ammonium Recovery

**DOI:** 10.3390/membranes12121233

**Published:** 2022-12-05

**Authors:** Hanna Kyllönen, Juha Heikkinen, Eliisa Järvelä, Antti Grönroos

**Affiliations:** VTT Technical Research Centre of Finland Ltd., P.O. Box 1000, FI-02044 VTT Espoo, Finland

**Keywords:** ammonium, nitrogen, membrane, recovery, scaling, precipitation, oxalic acid, sodium carbonate, mine water

## Abstract

One option for new nitrogen sources is industrial liquid side streams containing ammonium nitrogen (NH_4_-N). Unfortunately, NH_4_-N often exists in low concentrations in large water volumes. In order to achieve a highly concentrated NH_4_-Nsolution, scalant removal is needed. In this study, scalant removal by precipitation was investigated. At alkali pH, sodium carbonate (Na_2_CO_3_) was used as a precipitation chemical while at acidic pH, the chemical used was oxalic acid (C_2_H_2_O_4_). At alkali pH, high Na_2_CO_3_ dose was needed to achieve low content of calcium, which, with sulphate, formed the main scalant in the studied mine water. NH_4_-N at alkali pH was in the form of gaseous ammonia but it stayed well in the solution during pre-treatment for nanofiltration (NF) and reverse osmosis (RO). However, it was not rejected sufficiently, even via LG SW seawater RO membrane. At acidic pH with CaC_2_O_4_ precipitation, NF90 was able to be used for NH_4_-N concentration up to the volume reduction factor of 25. Then, NH_4_-N concentration increased from 0.17 g/L to 3 g/L. NF270 produced the best fluxes for acid pre-treated mine water, but NH_4_-N rejection was not adequate. NF90 membrane with mine water pre-treated using acid was successfully verified on a larger scale using the NF90-2540 spiral wound element.

## 1. Introduction

The ability to recover and reuse nutrients from wastewater could reduce dependency on energy-intensive processes used to synthesise mineral fertiliser products, such as nitrogen (N). N, a major component of fertilisers, is manufactured in the present day using a Haber–Bosch process, which accounts for 1–2% of worldwide energy demand, represents 3–5% of global annual natural gas consumption, and generates 4–8 tons of equivalent to carbon dioxide (CO_2_eq) per ton of N fertiliser per year [1,2,3]. On the other hand, through biological wastewater treatment, it is finally released back into the atmosphere as nitrogen gas (N_2_). The biochemical conversion of ammonium nitrogen (NH_4_-N) into N_2_ at the expense of energy, excludes the potential to recover and reuse N. To this end, more focus has been placed onto the recovery of N from liquid side streams by mature technologies, such as stripping, as well as novel technologies, such as reverse osmosis (RO) and electrodialysis (ED) [1].

Given the limited availability of water in many countries, water reuse in industry is increasing. Apart from that, industry can have a strong environmental impact on surrounding water faces. In the mining industry, most metals are obtained from ore bodies containing sulfidic minerals that are oxidised to sulphate (SO_4_) during the metal extraction process. Therefore, apart from metals, SO_4_ is a common impurity in mining waters and wastewaters of hydrometallurgical processing [4], referred to as mine waters in this study. Mine waters can also contain N as an impurity from incomplete detonation of N-rich explosives and from N-containing chemicals used in enrichment processes [5,6]. Environmental limits of impurities when discharging the mine water to surrounding water faces vary from site to site, but there is a trend towards tighter limits in the future. Meanwhile, there is significant potential for raw materials lurking in liquid side streams.

Recently, membrane filtration to purify liquid side streams in terms of different driving forces, such as those that are pressure-driven, osmosis-driven, thermally driven, or electrically driven, has received much attention. Membrane filtration, i.e., nanofiltration (NF) and RO, can be employed to produce high-quality water for reuse or discharge, and it has the advantage of low operating costs [7]. The target today is to gain as much purified water as possible, i.e., to obtain high water recovery (WR), and at the same time, membrane filtration generates impurities at high concentrations to be deposited or utilised. When aiming for high WR, the performance is mostly limited by membrane fouling and scaling. There is a great risk of calcium sulphate (CaSO_4_), i.e., gypsum scaling, when mine water is purified by pressure-driven NF or RO. Mine water contains originally a lot of SO_4_, and calcium (Ca) is added as a treatment chemical, thus scaling can occur on the membrane surface when sparingly soluble CaSO_4_ is concentrated beyond its solubility limit. This leads to significant flux reduction and salt rejection impairment and limits the WR of the desalination process to as low as 50–60% [8,9,10]. 

The use of antiscalants is a widely adopted technique to prevent scaling of calcium carbonate (CaCO_3_) and CaSO_4_. Additionally, pre-treatment of the feed by pH adjustment, ion exchange and chemical precipitation may help to reduce scaling [9]. Chemical precipitation is a process that uses specific chemicals to precipitate targeted sparingly soluble salts prior to clarification and pre-filtration to increase the WR of subsequent NF or RO. An alkaline solution, such as sodium carbonate (Na_2_CO_3_), soda ash (NaHCO_3_), or caustic soda (NaOH) with carbon dioxide (CO_2_), are the most studied chemicals for RO feed water to precipitate and reduce its hardness [7,11]. CaCO_3_ precipitation occurs when dissolved Ca and carbonate (CO_3_^2−^) are mixed at concentrations that exceed the solubility of CaCO_3_ [12]. The formation rate is controlled by many factors, such as super saturation, temperature and pH. An increase of solution pH significantly enhances CO_2_ gas absorption to solution, and influences ionisation [13]. At high pH, the CO_3_^2−^ form is dominant [14], which leads to the acceleration of CaCO_3_ nucleation and crystal growth. The increase in pH favours the formation of the most stable polymorph, calcite, instead of the more soluble, meta-stable vaterite [13,14,15]. 

Oxalic acid (C_2_H_2_O_4_) has long been noticed to precipitate Ca from such solutions as calcium oxalate (CaC_2_O_4_) [16]. CaH_2_O_4_ is a sparingly soluble salt, which has led to harm for industrial processes. Thus, degradation of C_2_H_2_O_4_ has been studied to avoid precipitation problems [17]. It has not, however, been used as a precipitation chemical for scaling prevention in membrane filtration. With it being more expensive than widely used inorganic precipitation chemicals, it has not been desirable for scaling control unless other benefits would have been obtained, such as the low pH requirement for the subsequent separation process. Low pH can be meaningful for having NH_4_-N in the solution instead of ammonia (NH_3_) gas. The NH_4_/NH_3_ dissociation equilibrium point is at pKa = 9.24 [18]. Above, the NH_4_/NH_3_ dissociation equilibrium point NH_4_-N is in NH_3_ form, which is a unionised and small volatile molecule, molecular weight (MW) of 17 Da, which readily diffuses through RO membranes. The retention of uncomplex NH_3_ can vary between 10% and 40% [19]. On the other hand, the high solubility of NH_3_ compared to that of other dissolved gases in water, like CO_2_ or oxygen (O_2_), and the very low value of the Henry constant may make it difficult to remove free NH_3_ from solutions [20]. 

NH_3_ volatilisation from the aqueous to the gas phase is commonly calculated using Henry’s law. One underlying assumption in these calculations is that Henrys’ law is valid for dilute aqueous systems, which, for NH_3,_ are concentrations up to 1000 mg/L [21]. The driving force for NH_3_ volatilisation from solution is normally considered to be the difference in NH_3_ partial pressure between that which is in equilibrium with the liquid phase and that which is in the ambient atmosphere. The equilibrium vapour pressure of NH_3_ is controlled by the concentration, which, in the absence of other ionic species, is affected by the NH_4_-N concentration and pH [22]. NH_3_ volatilisation increases linearly with the concentration, and curvilinearly with temperature and solution pH. At a given pH, the fraction of un-ionised NH_3_ in solution increases with temperature [21]. 

To produce a usable chemical or fertiliser of NH_4_-N from liquid side stream, NH_4_-N concentration needs to be high. Typically, commercial NH_3_ stripping systems can achieve ammonium sulphate ((NH_4_)_2_SO_4_) concentration between 25 and 40 m%, equivalent to 6–10% of N, which is marketed as a liquid fertiliser [20]. Hollow fibre liquid–liquid membrane contactor (LLMC) with gas-permeable membrane is an attractive technology for the recovery of volatile compounds, such as NH_3_, with a low level of impurities [23]. LLMCs have been postulated as an eco-friendly technology for NH_3_ recovery. The optimum conditions for reject water from wastewater treatment plants has been found to be pH 10 while it is obtaining a 4% (NH_4_)_2_SO_4_ solution [24]. N concentrations of 1.5–3% are required for liquid inorganic macronutrient fertilisers. 

In this study, a scalant-controlled concept for maximum NH_4_-N concentration of mine water from a gold mine was designed and experimentally studied. N existed in large volumes and in low concentrations, thus high WR and volume reduction factor (VRF) was required to obtain maximum NH_4_-N recovery. Mine water was characterised and based on scalant removal being carried out by precipitation at different pH levels. Clarification and microfiltration (MF) were used to remove precipitates before membrane filtration. The effect of pre-treatment on concentration of N, especially NH_4_-N, by means of NF and RO, was worked out. 

## 2. Materials and Methods

The separation of scalants for the recovery of NH_4_-N and nitrate nitrogen (NO_3_-N) was studied at laboratory scale, using mine water sample from a Finnish gold mine according to the scheme in Figure 1. The effect of pH was studied on N content during scalant removal by precipitation aided by clarification and MF, and membrane concentration using either NF or RO.

### 2.1. Characterisation

Received water samples as well as samples from precipitation and membrane filtration were characterised in terms of water quality and nutrient content. A Hach DR3900 laboratory spectrophotometer (Hach, Loveland, CO, USA) was used for the main analysis [25]. Chemical oxygen demand (COD) was analysed using the cuvette method LCK114, total nitrogen (N total) using the method LCK138, NH_4_-N using methods LCK302 and LCK303, NO_3_-N using the method LCK339, potassium (K) using the method LCK228, chloride (Cl) using the method LCK311, Ca and magnesium (Mg) using the method LCK327, as well as SO_4_ using the method LCK153 [25]. 

Conductivity was measured using a VWR Conductivity meter CO 3000 H (VWR, Germany). pH was measured using a VWR pH1000 pH meter (VWR, Darmstadt, Germany). Inductively coupled plasma-atomic emission spectroscopy (ICP-OES) was used with SFS-EN ISO 11885 to analyse aluminium (Al), Ca, copper (Cu), Mg, manganese (Mn), iron (Fe), nickel (Ni), chrome (Cr), phosphorous (P), barium (Ba), sulphur (S), sodium (Na), K, silicon (Si), and zinc (Zn). Osmotic pressure was measured using a Wescor Vapro^®^ Model 5600 Vapor Pressure Osmometer (Wescor Inc, South Logan, UT, USA).

### 2.2. Precipitation

Based on the mine water characterisation CaSO_4_, gypsum was the main scalant in the water. Ca removal prior to NF or RO was carried out using Na_2_CO_3_ and C_2_H_2_O_4_, both in powder form. The pH of water was increased to 10.1 with 4 g/L Na_2_CO_3_, and lowered to 2.2 with 0.77 g/L C_2_H_2_O_4_ in order to achieve sufficient Ca reduction. 

Precipitates were separated from the liquid using clarification and subsequent MF using Whatman ME25 filter paper. MF permeate for feed was used as such for NF and RO or pH was decreased by 2 M sulphuric acid (H_2_SO_4_) before NF and RO. The solids removal method after precipitation and prior to spiral wound membrane module was carried out by clarification overnight. 

### 2.3. Membrane Filtration

Studied NF membranes were NF270 (DuPont, Wilmington, DE, USA), and NF90 (DuPont), with molecular weight cut-offs (MWCO) of 400 Da and 200 Da, respectively. Studied RO membrane was LG SW (LG Chem, Seoul, Republic of Korea) which is suitable for high salinity seawater RO (SWRO) applications. 

The performance of different membranes was evaluated by conducting lab scale filtration tests using a dead-end filtration cell HP4750X (Sterlitech, Auburn, WA, USA), with magnetic agitation for increased shear on a membrane’s surface. Membranes were characterised using manufacturers’ specifications, testing salt rejections using 2 g/L MgSO_4_ for NF90 and NF270 with 4.8 bar pressure and 15% recovery, and 32 g/L NaCl for LG SW with 55 bar pressure and 8% recovery.

The used feed water volume was 100 mL, the temperature was 23 ± 1 °C (the room temperature), the operating pressure was 20–40 bar for NF and 40–80 bar SWRO, and the effective membrane area was 14.6 cm^2^. Argon gas was used for the pressurisation of the cell. 

Spiral wound membrane module NF90-2540 (DuPont) with membrane area of 2.6 m^2^ in a 40-bar steel pressure vessel (Sommer & Strassburger GmbH & Co. KG, Bretten, Germany) was used to verify NH_4_-N concentration on a larger scale. The membrane was characterised as lab scale NF90 membrane. 

## 3. Results

### 3.1. Liquids Characterisation

The mine water originally had clearly higher NH_4_-N content than NO_3_-N content, 170 mg/L and 13 mg/L, respectively (Table 1). NH_4_-N content was, however, too low for recovery using, e.g., MC technology, which requires at least NH_3_ 500 mg/L. Thus, the solution required concentration for subsequent recovery. In order to achieve the highest N content possible, it was essential to keep NH_4_-N content as high as possible during pre-treatment and subsequent NF and SWRO. Mine water had high Ca concentration, 390 mg/L, which formed, along with SO_4_, the main scalant, CaSO_4_, which was the possible hindrance for high WR and VRF. The concentrations of potential foulants Al and Fe were low enough for NF and SWRO. Mn concentration was higher than that typically guided by membrane manufacturers, 0.12 mg/L and less than 0.05 mg/L, respectively. Other metal concentrations were low in the studied mine water. 

### 3.2. Precipitation

Increasing the dosages of Na_2_CO_3_ or C_2_H_2_O_4_ produced lower Ca content in the solutions, as expected. A sufficiently low concentration of Ca was obtained with Na_2_CO_3_ when the dosage was as high as 4.0 g/L and the pH was 10.1 (Table 2). C_2_H_2_O_4_ produced somewhat higher final Ca content but the dosage needed for decreasing Ca was lower, 0.77 g/L, than with Na_2_CO_3_. C_2_H_2_O_4_ precipitation decreased the solution pH to 2.2. When Ca was reduced at a high pH of 10.1, there was a risk of losing NH_3_ as a gas from the liquid. Some decrease of NH_4_-N, maximum 18%, was observed during pre-treatment, but this was within the measurement accuracy.

Feed waters to be concentrated using NF and SWRO were those with the lowest Ca content in the solution (Table 3). Mg precipitated with Ca, especially when the pH was high enough for the formation of magnesium hydroxide. Additionally, K was a slightly lower for the precipitated samples compared to the feed with no precipitation. NH_4_-N as well as NO_3_-N and SO_4_ contents after pre-treatment were similar in each water sample. COD increased slightly due to usage of organic acid. Osmotic pressure was double its previous level after Na_2_CO_3_ treatment compared to the feed with no precipitation, thus some of the chemical remained in the solution. Osmotic pressure affects the VRF when filtering at the maximum pressure level of the membranes. 

### 3.3. Membrane Filtration

Salt rejections in membrane characterisation were 91.5–94.0% for NF270 and NF90 membranes and 92.9–93.0% for LG SW membrane based on conductivity measurements of feed and permeate.

During lab scale membrane filtration, the permeability of mine water filtration was measured as a function of VRF. Pre-treatment for the removal of the main scalant, i.e., CaSO_4_, did not improve the permeability of the filtrations, but both precipitations significantly assisted in an increase in VRF (Figure 2). C_2_H_2_O_4_-precipitated mine water produced higher VRF than Na_2_CO_3_ precipitated. On the other hand, the pH decrease in the feed water from 10.1 to 7.5 did not affect the permeability or VRF of precipitated mine water (Figure 3). The highest permeability was obtained using NF270 membrane. NF90 and LG SW produced the same permeability at pH 10.1, but the pressure needed for filtration when using LG SW was double that of NF90.

Although NF270 produced the highest permeability, NH_4_-N rejection at pH 2.2 with NF270 was not as good as with NF90 (Figure 4) due to differences in the MWCOs of the NF270 and NF90 membranes, 400 and 200 Da, respectively. This led to lower NH4-N concentration of NF270 concentrate than NF90 concentrate, 1.5 g/L and 3.0 g/L, respectively, although the VRF was the same, 25 (Table 4). The best rejection was achieved with NF90 with no precipitation. Then, the membrane scaling occurred and VRF was only 5, leading to NH_4_-N concentration of 0.8 g/L. At a high pH, 10.1, NH_3_ was not rejected sufficiently even by LG SW membrane. Above the NH_4_/NH_3_ dissociation equilibrium point pKa 9.24, NH_4_-N is in NH_3_ form, which as a small molecule readily diffused through SWRO membranes. pH adjustment from 10.1 to 7.5 before concentration increased the rejection near the level of C_2_H_2_O_4_ precipitation or no precipitation at all. NO_3_-N was not rejected well with NF membranes, but rejection with SWRO membrane was 84%. Ca concentration in some concentrates was lower than expected due to precipitation of the sample before analysis. 

Based on the obtained NH_4_-N rejection, VRF and energy consumption, NF90 and C_2_H_2_O_4_-precipitated mine water was selected for larger-scale verification studies using a NF 2540 spiral wound element. Verification filtration unfolded similarly to lab scale filtration using NF90 (Figure 5), but the achieved VRF at lower 20 bar pressure used was not as high as in lab filtration, 17 instead of 25. The NH_4_-N content in the concentrate was 1.9 g/L (Table 5). Obtained permeate quality was good regarding the compounds studied (Table 5 and Table 6), but the pH was low and would need to be increased when reused or discharged. 

Based on the mass and NH_4_-N content, NF90-2540 concentrate and permeate contained 96.3% and 3.7% of NH_4_-N, respectively. Similarly, in flat sheet tests, the division of NH_4_-N to concentrate and permeate was 93.3% and 6.7%, respectively.

## 4. Discussion

Mine water can be considered as a possible N-source for chemicals and fertilisers. N often exists in low concentrations in large water volumes. This is valid for mine waters from a gold mine, although feed water attempts to acquire as concentrated a stream as possible. In order to achieve concentrated NH_4_-N solution from mine waters and at the same time produce large volumes of purified water by membrane filtration, scaling removal is usually required in order to reach high WR and VRF. In mine waters, Ca often forms the main scalant with SO_4_, as it was with the studied feed water. Thus, removal of Ca is required, which can be carried out by precipitation with C_2_H_2_O_4_ as well as Na_2_CO_3_ chemicals. CaH_2_O_4_ and CaCO_3_ are both sparingly soluble Ca salts. 

To achieve low Ca content in the feed solution, a high dose of Na_2_CO_3_ may be required, which itself would produce a high-pH feed solution. NH_4_-N is in a gaseous form in water when pH is 10 or above. In this research the dosage was as high as 4.0 g/L and pH was 10.1. On the other hand, NH_3_ is a soluble gas and seemed to stay quite well in the solution. It stayed in the solution during precipitation, clarification, and subsequent MF, all of which were carried out as a pre-treatment method to NF and SWRO in this study. However, this research showed that the unionised and small NH_3_ molecule is not able to be concentrated even by SWRO membrane unless pH is decreased. When low pH is used for scalants removal with C_2_H_2_O_4_, or pH is decreased before concentration to nearly neutral, NF90 is able to concentrate NH_4_-N. NF270, which had larger MWCO than NF90, could clearly produce higher permeability than NF90, but NH_4_-N rejection, even at low pH, was not adequate. NF90 and LG SW produced the same permeability at pH 10.1, but the pressure needed for filtration when using SWRO was double that which was needed for NF.

Pre-treatment did not improve the permeability of the NF or SWRO, but both precipitations significantly assisted VRF increase. C_2_H_2_O_4_-precipitated mine water produced higher VRF than Na_2_CO_3_ precipitated. Due to high VRF, 25, and high rejection, 90%, NH_4_-N content in the NF90 concentrate was high at low pH. It was as its best in this research at 3.0 g/L, which would be high enough for the subsequent NH_3_ recovery, e.g., via stripping or MC technology. NO_3_-N was not rejected well with either of the NF membranes but rejection with the SWRO membrane was 84%. However, NO_3_-N concentration in the mine water was originally low. The quality of purified water from NH_4_-N concentration when using NF90 membrane with C_2_H_2_O_4_ pre-treatment was otherwise good, but the pH of purified water needs to be increased for water reuse or discharge. 

Mass balance of NH_4_-N from spiral wound membrane test resulted in 96% and 4% division of NH_4_-N to concentrate and permeate, respectively. Permeate amount was 96% of the feed with low concentration of NH_4_-N, i.e., 4.8 mg/L. In the future, industry can utilise the results for water reuse and minimising surface water uptake. The gold mine produces 70 m^3^/h of the studied mine water, from which it is possible to draw 66 m^3^/h of pure water with low NH_4_-N concentration. The developed concept is also usable by producing concentrated NH_4_-N stream in low volume. 

This study is part of circular economy development. When producing NH_4_-N concentrate from mine water, all the streams from the developed process need to be designed. Clarifier underflow from solids removal could go to a filter press, from which the filtrates from filtration and pressing stages could be returned back to the clarifier, and the filter cake is considered a suitable material for, e.g., mine backfilling or cement additive. Residual water from NH_4_-N recovery consists mainly of Na and SO_4_ and could be processed into usable chemicals, e.g., acid and base by electrodialysis (ED). These uses can be a part of future studies.

## 5. Conclusions

Mine water from a metal enrichment process featuring NH_4_-N was studied as a possible option for N recovery and water reuse. N existed in a low concentration in a large water volume and in order to achieve concentrated NH_4_-N solution, scalant removal by precipitation was needed. Precipitation was carried out at alkali pH 10.1 using Na_2_CO_3_ and at acidic pH 2.2 using C_2_H_2_O_4_ as a precipitation chemical, which both functioned well in removal.

Low concentration of Ca was obtained with Na_2_CO_3_ at high pH when N was in gaseous NH_3_ form. Instead, C_2_H_2_O_4_ precipitation produced a solution where N was in ionic NH_4_-N form. Soluble NH_3_ remained well in the solutions during pre-treatment with either of the chemicals but pH had a significant influence on the N rejection. Good NH_4_-N rejection was achieved at low pH using NF90, but at the high pH, gaseous NH_3_ was not rejected sufficiently even by means of an LG SW membrane. Pre-treatment did not improve the permeability of the NF or SWRO but both precipitations significantly assisted VRF increase. Good rejection, 90%, at acidic pH and high VRF, 25, when using NF90 membrane produced NH_4_-N concentration 3 g/L, which is a suitable concentration for subsequent N recovery. Concentration was successfully verified at pilot scale using NF90-2540 element.

## Figures and Tables

**Figure 1 membranes-12-01233-f001:**
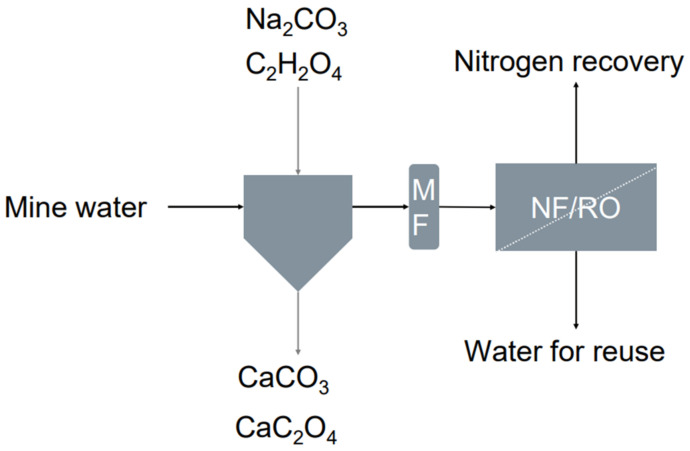
Studied process concept for nitrogen (N) concentration at a gold mine. Calcium (Ca) removal prior to nanofiltration (NF) or reverse osmosis (RO) was carried out using sodium carbonate (Na_2_CO_3_) and oxalic acid (C_2_H_2_O_4_). Ca was precipitated as calcium carbonate (CaCO_3_) and calcium oxalate (CaC_2_O_4_) and precipitates were removed by clarification and microfiltration (MF).

**Figure 2 membranes-12-01233-f002:**
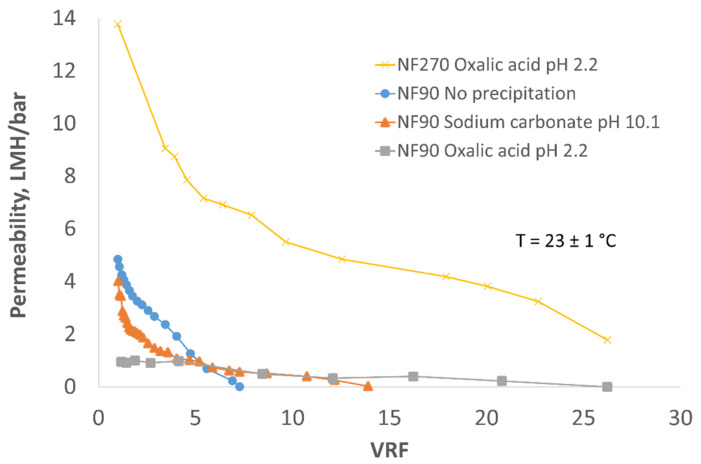
Concentration of pre-treated mine water using NF90 with no precipitation, precipitated using Na_2_CO_3_ and precipitated C_2_H_2_O_4_ compared to NF270 precipitated with C_2_H_2_O_4_.

**Figure 3 membranes-12-01233-f003:**
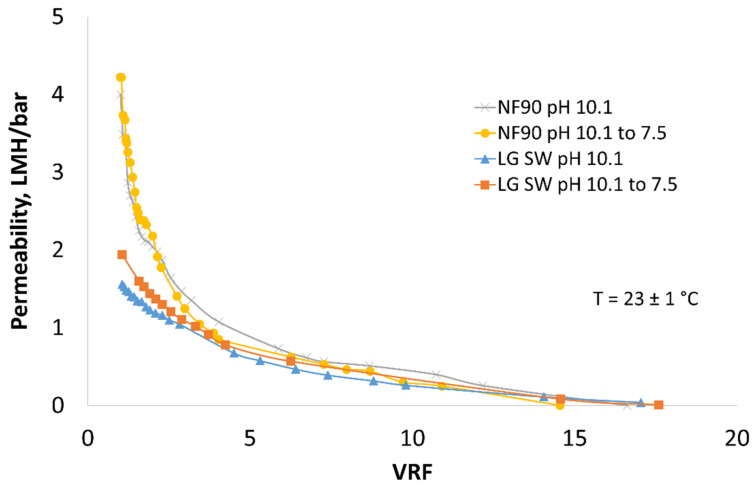
Concentration of pre-treated mine water using NF90, and LG SW precipitated using Na_2_CO_3_ without and with pH adjustment.

**Figure 4 membranes-12-01233-f004:**
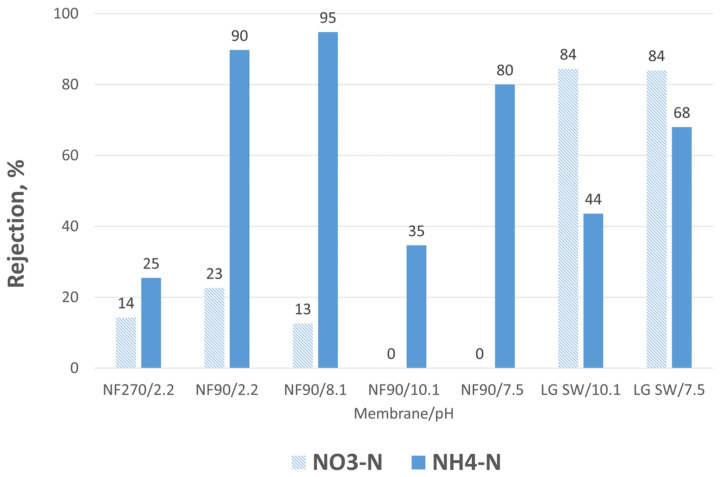
Nitrate nitrogen (NO_3_-N) and ammonium nitrogen (NH_4_-N) rejections using different pre-treatment and membranes.

**Figure 5 membranes-12-01233-f005:**
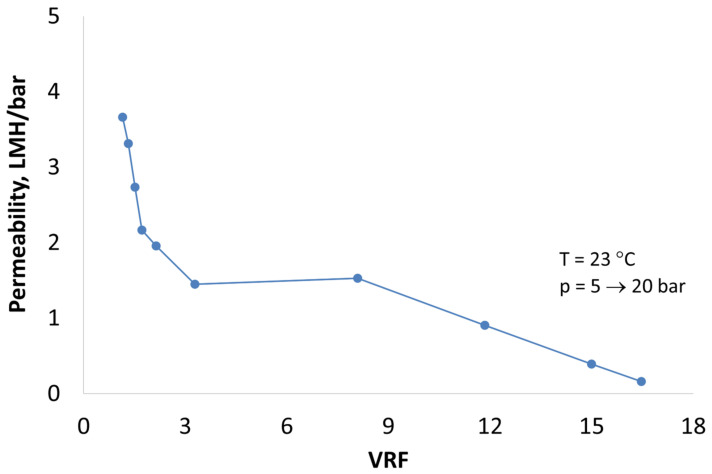
Permeability of C_2_H_2_O_4_-precipitated mine water when using NF90-2540 spiral wound element.

**Table 1 membranes-12-01233-t001:** Characteristics of mine water. Uncertainty of the measurements was 15–30%.

Parameter	Unit	Process Water
pH		7.1
Conductivity	mS/cm	6.1
Osmotic pressure	bar	1.6
Total-N	mg/L	174
NH_4_-N	mg/L	170
NO_3_-N	mg/L	13
Al	mg/L	<0.03
Ca	mg/L	390
Cu	mg/L	<0.003
Mg	mg/L	55
Mn	mg/L	0.12
Fe	mg/L	<0.05
Ni	mg/L	0.051
Cr	mg/L	<0.003
P	mg/L	<0.01
Ba	mg/L	0.007
S	mg/L	920
SO_4_	mg/L	2600
Na	mg/L	680
K	mg/L	36
Si	mg/L	8.2
Zn	mg/L	<0.005

**Table 2 membranes-12-01233-t002:** Precipitation of mine water sample. Concentrations were measured from the 0.45 µm filtrate.

Chemical	Dose g/L	pH	NH_4_-N mg/L	Ca mg/L	Mg mg/L
-	-	8.1	170	390	77
Na_2_CO_3_	0.8	8.3	140	170	59
Na_2_CO_3_	1.7	9.5	150	140	23
Na_2_CO_3_	4.0	10.1	150	7	34
C_2_H_2_O_4_	0.13	6.0	140	310	77
C_2_H_2_O_4_	0.77	2.2	170	81	56

**Table 3 membranes-12-01233-t003:** Feed water samples for nanofiltration (NF) and seawater reverse osmosis (SWRO).

Chemical	Dose g/L	pH	π Bar	NH_4_-N mg/L	NO_3_-N mg/L	Ca mg/L	Mg mg/L	SO_4_ mg/L	K mg/L	COD mg/L
-	-	8.1	1.6	170	13	390	77	2600	69	160
Na_2_CO_3_	4.0	10.1	3.5	150	12	7	34	2300	45	na
C_2_H_2_O_4_	0.77	2.2	2.0	170	12	81	56	2200	41	240

**Table 4 membranes-12-01233-t004:** Concentrations of feed, permeate and concentrates of the mine water when using the studied membranes.

Membrane/pH	VRF	Sample	NH_4_-N mg/L	NO_3_-N mg/L	Ca mg/L	Mg mg/L	SO_4_ mg/L	K mg/L
NF270/2.2		Feed	170	12	81	56	2200	41
	25	Concentrate	710	16	1000	1300	13,900	260
		Permeate	130	10	6	14	550	24
NF90/2.2		Feed	170	12	81	56	2200	41
	25	Concentrate	2950	15	620	1300	22,100	860
		Permeate	9	11	1	4	120	2
NF90/8.1		Feed	170	13	370	77	2600	69
No precipitation	7	Concentrate	770	27	1400	550	9500	200
		Permeate	18	9.8	0	3	33	0
NF90/10.1		Feed	150	12	7	34	2300	45
	17	Concentrate	370	40	100	630	24,000	780
		Permeate	80	12	0	5	30	1
NF90/7.5		Feed	130	12	18	36	4200	38
	14	Concentrate	1490	10	600	180	4400	510
		Permeate	25	12	0	0	15	6
LG SW/10.1		Feed	120	12	7	34	2300	45
	17	Concentrate	440	250	110	800	36,300	910
		Peremate	70	2	0	2	45	3
LG SW/7.5		Feed	130	12	18	36	4200	38
	17	Concentrate	2010	150	290	830	70,600	740
		Permeate	41	2	0	0	90	1

**Table 5 membranes-12-01233-t005:** Quality of C_2_H_2_O_4_-precipitated feed, permeate and concentrate in verification filtration using NF90-2540 element.

	pH	π mS/cm	NH_4_-N mg/L	NO_3_-N mg/L	Ca mg/L	K mg/L	Mg mg/L	Cl mg/L	SO_4_ mg/L	COD mg/L
Original	7.1	6.1	160	12.3	440	58	82	19	1900	160
Feed	2.1	8.4	160	13.8	110	73	110	21	2990	240
Concentrate	1.7	56	1910	15.7	540	660	980	20	36,300	110
Permeate	2.4	1.8	4.8	1.9	1	4	2	5.4	150	160

**Table 6 membranes-12-01233-t006:** Metal concentration of C_2_H_2_O_4_-precipitated feed and concentrate from verification filtration using spiral wound NF90-2540 element.

	Al mg/L	Ba mg/L	Cr mg/L	Cu mg/L	Fe mg/L	Mn mg/L	Na mg/L	Ni mg/L	P mg/L	Si mg/L	S mg/L	Zn mg/L
Feed	<0.03	0.007	<0.003	<0.003	<0.05	0.12	680	0.051	<0.01	8.2	920	<0.005
Concentrate	0.67	0.13	0.28	5.9	24	5.3	8800	1.2	<5	76	13,000	4.5

## Data Availability

The data presented in this study are available on request from the corresponding author.

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
