# Peer review of "Scalant Removal at Acidic pH for Maximum Ammonium Recovery"

_membranes, 2022, doi:10.3390/membranes12121233_

Round 1

Reviewer 1 Report

The study is well aligned with the journal's scope and boasts considerable novelty in its approach. However, the authors need to include the Conclusions in the manuscript. As it is missing in the manuscript. 

1.     Inclusion of a section named “Conclusion” clearly stating the conclusions derived from the study is significant.

2.     Future prospects and scope for industrial scale-up are paramount to highlight the possible avenues of further research on the topic

3.     Inclusion of diffusion data and its corresponding model equations are highly desirable for improving the quality of the publication.

4.     Figure 2 is incomplete.

5.     Instances of abbreviations are missing; consider using a nomenclature table for the same.

6.     Line 134 “cuvette method” – Including references highlighting the method's intricacies is highly advisable. Such details are crucial to alleviate the reproducibility of the research for further improvements

7.     Line 14-16: “Ammonia stayed well…LG SW membrane.”- Recommendation to elucidate briefly on the “LG SW membrane” and its significance in the manuscript.

8.     Line 81-82, 218: “It has, however, not to our knowledge to been used…membrane filtration.”- It is advised to change the phrasing for this sentence.

9.     Grammatical errors and the quality of write-up needs to be significantly revised by a native speaker.

The manuscript has to be Revised MAJORLY

Author Response

  1. Conclusions have been added.
  2. Future prospects have been added in Discussion: “Industry can in future utilize the results for water reuse and minimising surface water uptake. The gold mine produces 70 m3/h of the studied mine water, from which it is possible to draw 66 m3/h of pure water having low NH4-N concentration. The developed concept is also usable by producing concentrated NH4-N stream in low volume. “
  3. Diffusion data has been added to the Introduction: “Above, the NH4/NH3 dissociation equilibrium point NH4-N is in NH3 form, which is a unionised and small volatile molecule, molecular weight (MW) of 17 Da, which readily diffuses through RO membranes. The retention of uncomplex NH3 can vary between 10% and 40% [19].
  4. Figure 2 has been corrected.
  5. Abbreviations have been checked based on instruction of Membranes: “Acronyms/Abbreviations/Initialisms should be defined the first time they appear in each of three sections: the abstract; the main text; the first figure or table. When defined for the first time, the acronym/abbreviation/initialism should be added in parentheses after the written-out form.”
  6. Line 134, cuvette methods: A reference [25] has been added to Materials and methods. The reference has been added to the reference list “LCK Cuvette Test System | HACH. Available online: URL https://uk.hach.com/lck (accessed on 16 11 2022).
  7. LG SW membrane has been elucidated in Materials and methods: “Studied RO membrane was LG SW (LG Chem) which is suitable for high salinity seawater applications.”
  8. Line 81-82, 218 has been corrected: “It has not, however, been used as a precipitation chemical for scaling prevention in membrane filtration.”
  9. The manuscript has been revised by the native English-speaking person.

Reviewer 2 Report

In this study, a scalant controlled concept for maximum NH4-N concentration of gold mine wastewater was designed and experimentally studied. A system must be developed with a high concentration factor with good N-rejection must be achieved to achieve such an aim. The basic idea is original and has a high potential to be implemented and support the circular economy. However, some revisions are required before it can be considered for publication. few comments below can be addressed by the author to improve the manuscript quality.

The writing was probably done in hurry. Many typo errors were found. See below. Re-check the entire manuscript. Check line 13: what is "than?" Lines 31-32: what is the subject of the sentence?

Abstract: the objective, the scope, and the methodology of the research must be rewritten for better clarity. Currently, the abstract is hard to be understood.

Figure 1: X-axis title? It may be better if both membranes share the same y-axis.

The display of all figures must be improved.  

The conclusion section is missing.

Apart from showing the experimental results, authors need to perform mass-balance (I.e., Nitrogen) to confirm the findings can be acceptable.

Author Response

  • Typo errors have been corrected and the manuscript has been revised by the native English-speaking person.
  • Abstract has been rewritten.
  • Figure 1: name of X-axis is added. Same Y-axis has been used for NF90 and NF270.
  • Figures have been brought to manuscript by another way for improving the display of the figures.
  • Conclusion has been added.
  • Mass balance has been added to
    1. Results: “Based on the mass and NH4-N content NF90-2540 concentrate and permeate contained 96.3 % and 3.7 % of NH4-N respectively. Similarly, in flat sheet tests the division of NH4-N to concentrate and permeate was 93.3 % and 6.7 % respectively.”
    2. Conclusions: “Mass balance of NH4-N from spiral wound membrane test resulted in 96 % and 4 % division of NH4-N to concentrate and permeate respectively. Permeate amount was 96 % of the feed with low concentration of NH4-N, i.e. 4.8 mg/l.”

Round 2

Reviewer 1 Report

The comments are partially addressed. The manuscript can be accepted for publication

Author Response

Answers to the comments of Reviewer 2 (shown also to Reviewer 1): Conclusions is now in the manuscript after Discussion. Conclusions has been shortened and some sentences are moved to Discussion. Thus, Discussion is also somewhat revised. Additionally, Figures 2-5 are brought to manuscript as png-file to improve the quality of the figures.

Reviewer 2 Report

The manuscript has been substantially improved. However, some suggestions below are still required.
- The conclusion section is usually placed at the end, and not before the discussion.

- Many contents in the conclusion can be placed under the discussion section. Make the conclusion brief (100-150 words).
- I still suggest improving the quality of the figure display (Figs: 2-5).

Author Response

Conclusions is now in the manuscript after Discussion. Conclusions has been shortened and some sentences are moved to Discussion. Thus, Discussion is also somewhat revised. Additionally, Figures 2-5 are brought to manuscript as png-file to improve the quality of the figures.

Round 3

Reviewer 2 Report

All comments have been addressed.